# Cabs1 Maintains Structural Integrity of Mouse Sperm Flagella during Epididymal Transit of Sperm

**DOI:** 10.3390/ijms22020652

**Published:** 2021-01-11

**Authors:** Xiaoning Zhang, Wenwen Zhou, Peng Zhang, Fengxin Gao, Xiuling Zhao, Winnie Waichi Shum, Xuhui Zeng

**Affiliations:** 1Medical School, Institute of Reproductive Medicine, Nantong University, Nantong 226019, China; zhangxn@ntu.edu.cn (X.Z.); zww950820@163.com (W.Z.); zhaoxiuling@ntu.edu.cn (X.Z.); 2Institute of Life Science, Nanchang University, Nanchang 330031, China; zhangpeng_2021@163.com (P.Z.); gaofengxin@139.com (F.G.); 3School of Life Science and Technology, ShanghaiTech University, Shanghai 201210, China

**Keywords:** *Cabs1*, *AntiCabs1*, lncRNA, spermiogenesis, spermatogenesis

## Abstract

The calcium-binding protein spermatid-associated 1 (Cabs1) is a novel spermatid-specific protein. However, its function remains largely unknown. In this study, we found that a long noncoding RNA (lncRNA) transcripted from the *Cabs1* gene antisense, *AntiCabs1*, was also exclusively expressed in spermatids. *Cabs1* and *AntiCabs1* knockout mice were generated separately (using Clustered Regularly Interspaced Short Palindromic Repeat (CRISPR)-Cas9 methods) to investigate their functions in spermatogenesis. The genetic loss of *Cabs1* did not affect testicular and epididymal development; however, male mice exhibited significantly impaired sperm tail structure and subfertility. Ultrastructural analysis revealed defects in sperm flagellar differentiation leading to an abnormal annulus and disorganization of the midpiece–principal piece junction, which may explain the high proportion of sperm with a bent tail. Interestingly, the proportion of sperm with a bent tail increased during transit in the epididymis. Furthermore, Western blot and immunofluorescence analyses showed that a genetic loss of *Cabs1* decreased Septin 4 and Krt1 and increased cyclin Y-like 1 (Ccnyl1) levels compared with the wild type, suggesting that *Cabs1* deficiency disturbed the expression of cytoskeleton-related proteins. By contrast, *AntiCabs1*^−/−^ mice were indistinguishable from the wild type regarding testicular and epididymal development, sperm morphology, concentration and motility, and male fertility. This study demonstrates that Cabs1 is an important component of the sperm annulus essential for proper sperm tail assembly and motility.

## 1. Introduction

Spermatogenesis is a highly conserved cyclic process in which spermatogonial stem cells generate mature spermatozoa. It is precisely controlled by many genes and intrinsic signals. Many endogenous or exogenous factors may cause abnormal spermatogenesis and male infertility [1,2]. Spermiogenesis is the terminal phase of spermatogenesis and consists of the differentiation of newly formed spermatids into spermatozoa. During spermiogenesis, round spermatids undergo a drastic restructuring program, including elongation, condensation of the nucleus, and the formation of sperm acrosome and tail. Understanding the intricate mechanisms that control spermiogenesis has important implications for human health and reproduction. Additionally, further plasticity and maturation of sperm in the epididymis are essential for male reproduction. Although extensive studies suggest that these processes are elaborately regulated by many male germ cell-specific genes [3,4,5,6], the underlying mechanisms and novel pathogenic genes need to be further addressed. The identification of functional genes will provide novel avenues for genetic diagnosis and new insights into human infertility.

The calcium-binding protein spermatid-associated 1 (Cabs1) was identified as a novel calcium-binding protein specifically expressed in elongated spermatids in mice [7]. Recently, it was also found to be exclusively expressed or enriched in elongated spermatids of human, porcine, and rat testes [8,9,10]. The *Cabs1* gene was found to be present solely in mammalian genomes, and the full-length murine Cabs1 protein is composed of 391 amino acids and highly conserved, sharing 61% and 80% identical amino acids with human and rat Cabs1, respectively. The Cabs1 protein is localized in the acrosome and principal piece of the flagellum of mature sperm in the cauda epididymis. An in vitro study suggested that porcine Cabs1 may be involved in the acrosome reaction by controlling calcium signaling [8]. Cabs1 can be phosphorylated by a serine-threonine kinase (Casein kinase 2) that caused sperm deformities [10,11]. Furthermore, it might have some target sites of proteolytic enzymes. These imply that Cabs1 may be involved in sperm function regulation [12]. The calcium-binding properties of Cabs1 are well-established. A very recent study just showed that Cabs1-deficient mice are fertile to a certain extent [13]. However, the further physiological roles of Cabs1 in the male reproductive system have not been elucidated.

Long noncoding RNAs (lncRNAs), defined as ncRNAs longer than 200 nucleotides in length, do not encode proteins. It has been shown that lncRNAs are involved in many pathophysiological processes at the transcriptional and post-transcriptional levels, functioning as signaling, guiding, sequestering, or scaffolding molecules [14,15]. Numerous lncRNAs are highly enriched or exclusively expressed in testes and/or spermatogenic cells [16,17]. Some testes-specific lncRNAs regulate spermatogenesis, and their deficiency can cause male infertility in Drosophila [18]. However, deficiencies of many individual lncRNAs have no detectable impact on spermatogenesis in mice and zebrafish models [19,20,21]. The function of many lncRNAs in male reproduction remains unknown, especially in mammals, and further research is required. LncRNAs regulate their target genes in a cis manner when target genes are restricted to the same chromosome from which the lncRNAs are transcribed [22]. In this study, we identified a novel lncRNA, *AntiCabs1*, which was transcripted from the antisense of the Cabs1 gene.

Considering the known roles of *Cabs1* and lncRNAs, we predicted that *Cabs1* and *AntiCabs1* may play an important role in spermiogenesis and sperm maturation. We generated *Cabs1* and *AntiCabs1* knockout (KO) mice, using Clustered Regularly Interspaced Short Palindromic Repeat (CRISPR)-Cas9 technology, to investigate their physiological role in spermatogenesis and to examine whether *Cabs1* expression was regulated by *AntiCabs1* in a cis manner.

## 2. Results

### 2.1. Temporal and Spatial Expression Profiles of Cabs1 and AntiCabs1

The distribution of mouse *Cabs1* and *AntiCabs1* in various tissues was detected via RT-PCR. The results showed that *Cabs1* and *AntiCabs1* were exclusively expressed in mouse testis (Figure 1A and Appendix A). Consistent with mRNA expression, the Cabs1 protein was also specifically expressed in the testis (Figure 1B). The expression levels of both the *Cabs1* mRNA and protein sharply increased in postnatal four-week-old testes, during the elongating spermatid stage, and the expression was detected from the elongated spermatid (postnatal day 28) to adult stages (Figure 1C,D). Cabs1 was also detected in the caput and cauda epididymis, vas deferens, and mature sperm (Figure 1E). An immunofluorescence analysis showed that Cabs1 was mainly located in elongating spermatids and the acrosome and principal piece in mature sperm (Figure 1F). Taken together, these results demonstrated that *Cabs1* expression was spermatid-specific and may be important for spermiogenesis and sperm function regulation.

### 2.2. Generation of Cabs1 and AntiCabs1 KO Mice

To gain further insights into the function of *Cabs1* and *AntiCabs1*, we generated *AntiCabs1* and *Cabs1* KO mice using the CRISPR-Cas9 method (Figure 2A). Single genomic RNA (gRNA) was used to cut the genomic DNA of exon1 of the *Cabs1* gene, resulting in a decoding frame shift mutant of Cabs1. Western blot and immunofluorescence analysis confirmed the loss of the Cabs1 protein in *Cabs1*^−/−^ mice (Figure 2B–D). To avoid the potential effect of *AntiCabs1* deficiency upon *Cabs1* expression, we deleted the promoter region of *AntiCabs1*, which blocked *AntiCabs1* transcription (Figure 2A). Genotyping and RT-PCR data confirmed the generation of *AntiCabs1*^−/−^ mice (Appendix A). Two specific PCR products, specific 1 (950 bp) and specific 2 (280 bp), were amplified in *AntiCabs1*^+/+^ and *AntiCabs1*^+/−^ mice but not in *AntiCabs1*^−/−^ mice (Appendix A).

### 2.3. Genetic Loss of Cabs1 Caused Sperm Malformation

In *Cabs1*^−/−^ mice, there were no discernible changes in the development, morphology, and weight of the testis and epididymis compared with wild-type (WT) mice (Figure 3A–C). However, the genetic loss of *Cabs1* markedly affected sperm morphology and motility. As shown in Figure 3D, many spermatozoa from *Cabs1*^−/−^ mice exhibited several defects, such as a bent tail or a thinning of the annulus (ring-like structure in the tail region of sperm) region. These abnormal structures were further confirmed by transmission electron microscopy (Figure 3E). The annulus was easily identified in WT sperm as an hourglass-shaped structure at the midpiece boundary. Strikingly, the annulus was completely absent in mutant sperm, as shown in Figure 3D,E. For *AntiCabs1*^−/−^ mice, the annulus developed normally, and the morphology of the testis and epididymis (Appendix A), testis weight (Appendix A), and the concentration (Appendix A) and motility (Appendix A) of the sperm and male fertility (Appendix A) all showed normal phenotypes. Hence, *Cabs1*, rather than *AntiCabs1*, exerted an organizational role in the annulus for the proper architecture of the tail.

### 2.4. Defects in Sperm Flagella Occurred in the Epididymis of Cabs1^−/−^ Mice

To identify the site of onset of the structural defects observed in sperm from *Cabs1*^−/−^ mice, we performed a microscopic analysis of sperm collected from testes and the caput, corpus, and cauda epididymal regions in 430 and 310-mOsm high-salt (HS) buffer (Figure 4A–D). In testes, we observed some thinning of sperm flagella but only rare cases of angulation (Figure 4A). There was a progressive increase in the proportion of angulated sperm from the caput to the cauda epididymis, independently of the osmotic pressure change (Figure 4A–C), suggesting that angulation was caused during the normal transit of spermatozoa in the epididymis. However, the percentage of sperm with a thin annulus was higher in the caput than cauda epididymis in *Cabs1*^−/−^ mice (Figure 4E,F). There was a marked difference in annulus defects of sperm in the epididymis caput from fertile compared with infertile *Cabs1*^−/−^ mice (Figure 4E). Interestingly, this difference disappeared in sperm from the epididymis cauda (Figure 4F). Similar results were observed for the total abnormal sperm rate (Figure 4G,H). Collectively, we proposed that defects in sperm flagella of *Cabs1*^−/−^ mice mainly occurred in the normal transit process in the epididymis.

Although *Cabs1*^−/−^ mice had normal sperm numbers, the sperm total motility was significantly lower than WT mice (Figure 5A,B). Fertility tests by breeding WT and *Cabs1*^−/−^ adult males with WT adult females over a six-month period revealed that some *Cabs1*^−/−^ male mice (~30%) were sterile (Figure 5C). Interestingly, for fertile *Cabs1*^−/−^ male mice, their litter sizes were equivalent to those of WT mice (Figure 5D).

### 2.5. Deletion of Cabs1 Causes Abnormal Expression of Flagella-Related Proteins

Considering that AntiCabs1 was a lncRNA transcripted from *Cabs1* antisense, we constructed *AntiCabs1* and *Cabs1* overexpression H293 cell lines to determine whether *AntiCabs1* overexpression affected *Cabs1* expression (Appendix A). Overexpression of *AntiCabs1* by transfection with a lentiviral vector had no effect on the *Cabs1* expression in vitro (Appendix A). In *AntiCabs1* KO mice, the *Cabs1* mRNA and protein expression were equivalent to levels found in WT mice (Appendix A). Therefore, the *Cabs1* expression levels were not regulated by *AntiCabs1*.

To further study the underlying mechanisms that induced the sperm malformation in *Cabs1* KO mice, we investigated the expression of cytoskeleton-related proteins. Cyclin Y-like 1 (Ccnyl1) and septin 4 (Sept4) are flagella assembly-related proteins whose defects lead to similar phenotypes in sperm morphology. The *Ccnyl1* mRNA level was elevated; however, the *Sept4* and *Krt1* mRNA levels were lowered in Cabs1 KO mouse sperm (Figure 6A). Similarly, a Western blot analysis revealed that the expression of Krt1 and Sept4 were significantly decreased, but Ccnyl1 was increased, in mature sperm of *Cabs1*^−/−^ compared with WT mice (Figure 6B,C). The immunofluorescence results further confirmed that the expression levels of Ccnyl1, Sept4, and Krt1 were changed (Figure 7). In addition, their expression patterns or locations in sperm of *Cabs1*^−/−^ mice were also disturbed (Figure 7, red arrows). These results suggested that the genetic loss of *Cabs1* impaired normal sperm formation by disturbing cytoskeleton-related protein expression essential for normal flagella assembly.

## 3. Discussion

By means of gene knockout technology and animal models, many male germ cell-specific genes or gene isoforms were identified that play essential roles in sperm formation [23]. Using a similar approach in the present study, we revealed that Cabs1 is required for maintaining the structural integrity of sperm flagella. Our results indicated that murine Cabs1 was located in the acrosome of mature sperm, as well as confirming the previously described expression in elongating and elongated spermatids and the principal piece of sperm flagellum [7]. The acrosome expression was consistent with the expression reported in porcine sperm [9], suggesting that mouse Cabs1 is involved in sperm formation, as well as mature sperm function, involving the regulation of the acrosome reaction and sperm motility. Data from a rat study showed the presence of Cabs1 in the mitochondrial inner membrane in sperm [19]. In humans, CABS1 was found in salivary glands and the testis and was associated with stress and anti-inflammation [24,25]. However, this anti-inflammatory domain was only found in primates of the infraorder Simiiformes [12]. In the present study, we also found that *Cabs1* mRNA was expressed in the prostate and small intestine in humans (Appendix A). Overall, the expression of Cabs1 in the reproductive system indicated that it may have a conserved function in male fertility.

In this study, *AntiCabs1* and *Cabs1* KO mice were generated to investigate the function and regulation of *Cabs1* in male reproduction. Antisense lncRNA can regulate its neighborhood coding gene in a cis manner [26]. Many lncRNAs are derived from bidirectional promoters or overlap with promoters, regulatory regions, or exons of sense or antisense genes [27]. Therefore, the targeted deletion of an intact lncRNA may influence overlapping genes in the same loci. Hence, we selectively deleted the *AntiCabs1* promoter to block transcription of this lncRNA while theoretically maintaining the structures of the neighboring genes [28]. Our genomic and Western blot data demonstrated the successful generation of *Cabs1* and *AntiCabs1* KO mice. A reproductive phenotype analysis revealed that *Cabs1* deletion led to abnormal sperm with a bent tail and thinning of the annulus region, suggesting that Cabs1 was required for structural assembly of the sperm tail. The teratospermia may contribute to the impaired fertility observed for *Cabs1* KO males. Interestingly, some males were sterile, but others remained fertile. The mating results indicated that epididymal and ejaculated spermatozoa may have different fertility properties. Our findings were consistent with a recent study that *Cabs1* KO mice were still sterile but the number of pups and litters were significant less than WT controls [13]. However, no other reproductive phenotypes were reported in their study. Many gene mutations, such as the disruption of CatSper channel member genes, do not alter the appearance of sperm but cause male fertility due to sperm dysfunction [29]. Hence, we propose that the sperm with a normal appearance may be dysfunctional in some way in *Cabs1* KO infertile mice. As abnormal sperm is about 60% and the other sperm appear to be normal in morphology and motility, some of KO mice are fertile. Since the Cabs1 protein was also located in the acrosome, and it has been reported to be involved in the acrosome reaction and phosphorylation regulation [8,10], hence, for those sterile male mice, sperm functions such as capacitation and the acrosome reaction might be dysregulated, which reached a maximal threshold value, resulting in male infertility. In addition, a disrupted capacity of Ca^2+^, Mg^2+^, and Zn^2+^ of Cabs1 may, to some extent, explain the infertility in *Cabs1* KO mice [3,7], but this should be further investigated.

For the *AntiCabs1* KO mice, there were no appreciable phenotypic changes. Recent studies on elegans, mice, and zebrafish also showed that many individual lncRNAs were dispensable for spermatogenesis and fertility [19,20,30,31]. *AntiCabs1* may be compensated by other lncRNAs, because there are abundant lncRNAs in cells at the spermiogenesis stage [32]. The *AntiCabs1* may have redundant, subtle, or context-dependent roles, which need to be addressed in future studies. Our results showed that *AntiCabs1* gain-of-function in vitro, and loss-of-function in vivo, did not affect *Cabs1* expression at the molecular level (Appendix A), suggesting that *AntiCabs1* may be superfluous and have overlapping functional roles in mechanisms such as the scaffold of protein complexes during the process of normal sperm formation.

Sperm are exposed to a higher physiological hypertonic stress (~430 mOsm) in the epididymis than in regular HS and phosphate buffer saline buffers and the female genital tract (~310 mOsm). The deletion of some genes will lead to abnormal sperm with a bent tail, which result from the interference with osmoregulation in epididymis or mediums rather than the defect of spermiogenesis or maturation [33,34]. To exclude the influence of osmotic pressure changes on sperm deformity [34,35], we investigated the ratio of abnormal sperm in 310 and 430-mOsm HS buffer. Our data showed that the number of sperm with a bent tail from the caput epididymis was higher in 310 mOsm than in 430 mOsm HS buffer. It is possible that immature sperm in *Cabs1* KO mice were more sensitive to changes in the osmotic pressure. The progressive increase in the proportion of angulated sperm from the caput to cauda epididymis, independently of osmotic pressure changes, indicated that the angulation of sperm occurred during the normal transit of spermatozoa in the epididymis, possibly reflecting a cumulative effect over time or stress from microenvironmental changes such as ion concentrations or secreted protein from the caput to cauda epididymitis [36,37]. Additionally, although the total abnormal sperm rate was higher in the caput epididymis, no statistical difference was found in sperm from the cauda epididymis in infertile compared with fertile *Cabs1* KO mice. Sperm have many protein exchanges within the epididymis. Many proteins are known to be added to or removed from sperm by exosomes to promote sperm maturation [36,38,39]. Therefore, in *Cabs1* KO mice, some sperm with a thin annulus may have their morphology restored during the epididymal transit process.

The disruption of several testis-specific or enriched genes resulted in similar phenotypes exhibited by *Cabs1* KO mice. For example, spermatozoa from *Sept4*^−/−^, *Sepp1*^−/−^, *Krt1*^−/−^, and *Ccnyl1*^−/−^ mice showed a thinning annulus and a bent tail but also showed other defects in mitochondrial or capacitation functions [40,41,42,43,44]. These proteins, along with Cabs1, are essential for sperm flagellum assembly, especially for the annulus formation. Interestingly, these male KO mice were fully sterile, whereas *Cabs1*^−/−-^ male mice exhibited subfertility. Female *Cabs1*^−/−^ mice displayed normal fertility, consistent with the fact that Cabs1 is a male germ cell-specific gene and not expressed in females. Cytoskeleton-related proteins such as actin and Krt1 were thought to be involved in the cytoskeletal assembly of sperm flagellum [45,46]. To uncover the potential underlying mechanism that induced sperm abnormality and impaired fertility in *Cabs1* KO males, Ccnyl1, Sept4, and other cytoskeleton-related protein Krt1 expression levels were investigated (Figure 6 and Figure 7). We found that Ccnyl1 expression was elevated; however, Sept4 and Krt1 were reduced in sperm from cauda epididymitis, indicating that the expression of flagella-related proteins was disturbed in *Cabs1* KO mice. All these indispensable proteins for the annulus and tail assembly might form a complex or affect each other. Cabs1 deficiency lead to changes of the Sept4 and Krt1 expression levels and locations, which might contribute to sperm malformation and even subfertility due to the sperm tail structure being easily damaged. The upregulated proteins may be compensation for the loss of *Cabs1*. However, the underlying mechanisms need to be further explored. For example, why are only some *Cabs1* KO male mice fertile? How does Cabs1 disruption affect those flagella-related proteins and lead to sperm malformation?

Taken together, our findings demonstrate that the lack of *Cabs1* causes sperm tail deformation during the normal transit of spermatozoa in the epididymis. The assembling proteins of flagella may be involved in this process. Ongoing efforts to define the function of Casb1 will be important for the identification of novel causative genes for human infertility.

## 4. Materials and Methods

### 4.1. Animals

The CRISPR-Cas9 system was employed to generate *Cabs1* and *AntiCabs1* KO C57/BL6 mice. Briefly, the gRNA (5′-AGTAATTGACCCCTTAGGTTTGG-3′) to mouse *Cabs1* gene and Cas9 mRNA were co-injected into fertilized mouse eggs to generate targeted KO offspring. F0 founder animals were identified by PCR followed by sequence analysis, which were bred to WT mice to test germ line transmission and F1 animal generation. For *AntiCabs1* (NONCODE GENE ID: NONMMUG032913.2), two gRNAs (gRNA1: 5′-AACTACAAATACCCATGGATAGG-3′ and gRNA2: 5′-AAATTCAGGATATTAATCTCAGG-3′) were designed to delete the promoter region of *AntiCabs1*. After the mutant *AntiCabs1* mice were established, genotyping was performed by multiplex PCR. The WT allele generated a band at 498 bp, while the KO allele was at 760 bp. *Cabs1* KO mice were validated by the Sanger sequencing (Appendix A). One line of KO mice was retained for the next experiment, because 3-line KO mice have similar phenotypes in sperm morphology. Animal genotyping was performed by PCR with genomic DNA isolated from tail clips of mice. The PCR primers are listed in Appendix A. For a fertility test, 2-month-old KO and WT male mice were mated with proven fertile females (one male/one female per cage) for 3 months. The number of pups produced from each female was recorded and analyzed. Mice were treated in accordance with the guidelines approved by the Institutional Animal Ethical Committee (IAEC) of Nantong University (SYXK (Su) 2017-0046, July 2017, Nantong, China).

### 4.2. Histological Analysis 

The WT and KO mouse testis and epididymis of *Cabs1* and *AntiCabs1* were removed and immediately fixed in 4% formaldehyde solution, embedded in paraffin, sectioned at 5 μm, and stained with hematoxylin and eosin (H&E) using standard procedures. Pathological changes were observed and photographed under a light microscopy (Olympus Corp., Center Valley, PA, USA).

### 4.3. Sperm Morphology and Motility Analysis

Sperm cells collected from the caput, corpus, and cauda of epididymites were suspended in HS (high-saline; 135-mM NaCl, 5-mM KCl, 1-mM MgSO_4_, 2-mM CaCl_2_, 20-mM HEPES, 5-mM glucose, 10-mM lactic acid, and 1-mM Na-pyruvate at pH 7.4 with NaOH) buffer at 37 °C. Sperm were captured under a microscope to analyze the morphology. Sperm motility was analyzed using a computer-assisted sperm analysis (CASA) system (Hamilton Thorne, Inc., Beverly, MA, USA). At least 200 cells from each sample were counted for statistical analysis.

### 4.4. RT-PCR and Real-Time Quantitative PCR (qPCR)

Total RNA was isolated using RNAiso Plus (Takara, Beijing, China) according to the manufacturer’s instructions and dissolved in RNase-free water. The qualified RNA was reverse-transcribed to cDNA using a PrimeScript™ RT reagent Kit with gDNA Eraser (Takara). PCR amplification of *Cabs1* and *AntiCabs1* transcripts was conducted using Premix Taq™ (Takara Taq™ Version 2.0 plus dye, Takara) according to the manufacturer’s protocols. PCR products were separated by electrophoresis on a 1.5% agarose gel and visualized by ethidium bromide staining. qPCR was performed by SYBR^®^ Premix DimerEraser™ (Takara) with an ABI StepOnePlus real-time PCR system (Applied Biosystems, Foster City, CA, USA), with primers specific for mouse *Cabs1* mRNA. The 2^−^^△△Ct^ quantification method was utilized to detect the gene expression level. The PCR assay was as follows: 98 °C for 2 min, followed by 40 cycles of 98 °C for 10 s, 60 °C for 30 s, and 72 °C for 30 s, with a final extension at 72 °C for 2 min. *β-actin* was employed as an endogenous control to normalize the gene expression. The primer sequences used for qPCR were shown in Appendix A. All primers were designed with National Center for Biotechnology Information Primer-Blast tool and synthesized by Genewiz, Inc. (Genewiz, Suzhou, China).

### 4.5. Western Blot and Immunofluorescence Staining

Protein samples of cells or tissues isolated with radio immunoprecipitation assay (RIPA) buffer (Beyotime Biotechnology, Shanghai, China) were separated on SDS-PAGE gels and transferred to polyvinylidene fluoride (PVDF) membranes. The membranes were blocked in 5% BSA for 1 to 2 h at room temperature and probed with primary antibodies as follows: Cabs1 (produced by ourselves), Ccnyl1 (Bioss, Beijing, China), Sept4 (Bioss and ABclonal, Beijing and Wuhan, China), Krt1 (ProteinTech and ABclonal, Wuhan, China), GAPDH (Sigma, St. Louis, MO, USA), and β-actin (Sigma) antibodies overnight at 4 °C. The PVDF membranes were incubated with secondary antibodies conjugated to horseradish peroxidase at a dilution of 1:5000 (Santa Cruz Biotechnology, Dallas, TX, USA) and visualized by chemiluminescence detection (Thermo Scientific, Rockford, IL, USA) and quantitated using Quantity One Software (Bio-Rad, Hercules, CA, USA). For immunohistochemistry, mouse testis was fixed in 4% paraformaldehyde and embedded in paraffin. Sections (5 μm) were deparaffinized, perforated, blocked (3% BSA), and then incubated with Cabs1 antibody overnight at 4 °C. The fixed sperm cells with 4% paraformaldehyde were adhered on a culture dish and incubated with Cabs1 antibody overnight at 4 °C, followed by incubation for 1 h with DyLight 488 anti-rabbit secondary antibody (Thermo Fisher Scientific, Carlsbad, CA, USA) at room temperature, then co-labeled with DAPI (Sigma). Photomicrographs were taken with an Olympus FV1000-IX81 confocal laser scanning biological microscope (Olympus, Tokyo, Japan).

### 4.6. Transmission Electron Microscopy (TEM)

Sperm were fixed in 3% glutaraldehyde and 1.5% paraformaldehyde (Sigma) overnight at 4 °C. After a few washes with 0.1-M phosphate buffer, the spermatozoa were fixed with 2% osmic acid, washed, and dehydrated through an ethanol series and embedded in Epon812 resin. Thin sections were stained with uranyl acetate and lead citrate for 5 min. Images were acquired from a TEM (JEM-1200EX, Japan Electronics Co., Ltd., Tokyo, Japan).

### 4.7. Cell Culture and Transfection

The H293 cells were cultured in Dulbecco’s modified Eagle’s medium (DMEM) (Gibco, Grand Island, NY, USA), supplemented with 10% fetal bovine serum and 1% penicillin/streptomycin (Penn/Strep), and maintained in a humidified atmosphere containing 95% air and 5% CO_2_ at 37 °C. Lentivirus vectors, pLV(Exp)-Enhanced Green Fluorescent Protein (EGFP):T2A:Puro-EF1A>HA/mCabs1 and pLV(Exp)-EGFP:T2A:Puro-EF1A>AntiCabs1 (Appendix A), were constructed and packed by Cyagen US Inc. (Guangzhou, China). pLV(Exp)-EGFP:T2A:Puro-Null was used as a control vector. The *Cabs1* and *AntiCabs1* overexpression cells were verified by the EGFP protein expression with confocal microscopy (Appendix A) and by the specific PCR product amplifications (Appendix A). Then, the AntiCabs1 vector was transfected into Cabs1 overexpressed cells for 60, 72, and 96 h to address the effect of *AntiCabs1* on *Cabs1* expression. The protein expression was detected by Western blot, as mentioned above.

### 4.8. Statistical Analysis

Results are expressed as the mean ± standard error (SE) of the mean. All statistical analyses were performed using the statistical software GraphPad Prism (version 5.01, GraphPad Software, San Diego, CA, USA). *p* < 0.05 was regarded as statistically significant, and *p* < 0.01 was considered extremely statistically significant. Differences between the controls and the different samples were assessed with Student’s *t*-tests.

## Figures and Tables

**Figure 1 ijms-22-00652-f001:**
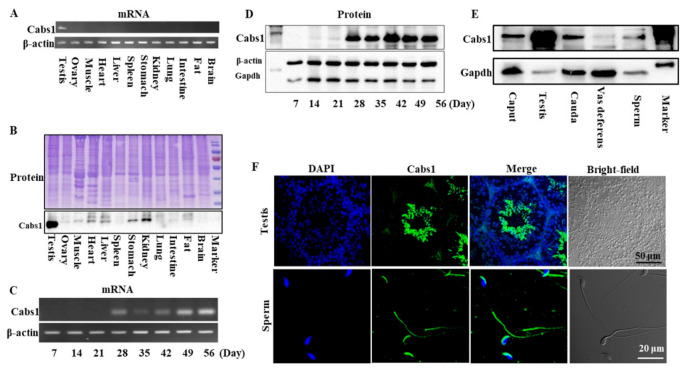
Temporal and spatial expression profiles of calcium-binding protein spermatid-associated 1 (Cabs1). (**A**) The mRNA and (**B**) protein expression of *Cabs1* in the testis, ovary, muscle, heart, liver, spleen, kidney, lung, intestine, fat, and brain. (**C**) mRNA and (**D**) protein expression of *Cabs1* in the testis at different stages of development. Total RNA was isolated from samples using a RNAiso Plus kit, and RT-PCR was performed to detect *Cabs1* expression levels. *β-actin* gene expression was used as a loading control. (**E**) Protein expression of *Cabs1* in the caput and cauda epididymis, vas deferens, and mature sperm. Equal amounts of total protein were subjected to SDS-PAGE and immunoblot analysis with antibodies directed against Cabs1, β-actin, and Gapdh. The blots are representative of three independent experiments. β-actin and Gapdh were used as the internal references. (**F**) Localization of the Cabs1 protein in mouse testis and sperm. Immunofluorescence labeling of frozen sections of mouse testis and mature sperm with the Cabs1 antibody and nuclear staining with DAPI (4′,6-Diamidino-2-Phenylindole, Dihydrochloride). Imaging was performed on a confocal microscope.

**Figure 2 ijms-22-00652-f002:**
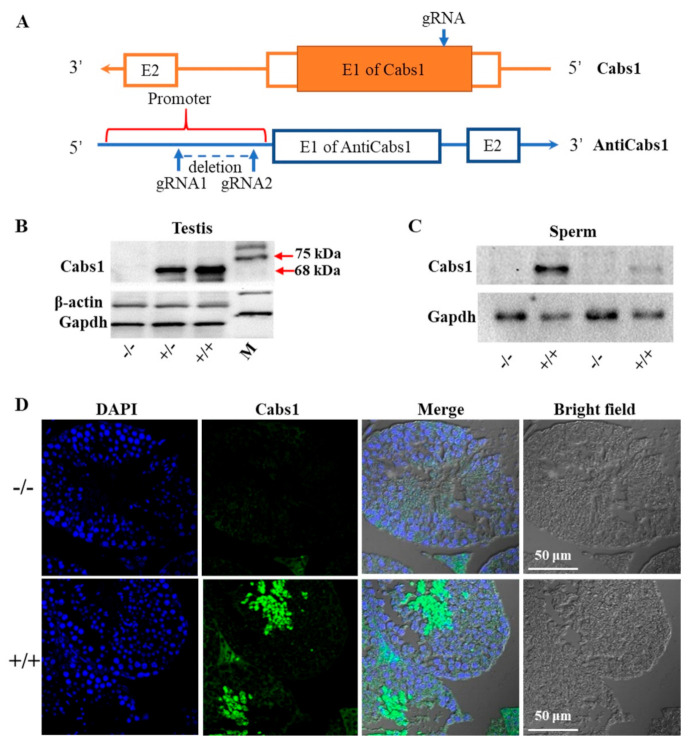
Generation of calcium-binding protein spermatid-associated 1 (Cabs1) and antisense of the *Cabs1* gene (*AntiCabs1*) knockout (KO) mice. (**A**) A schematic illustration of the Clustered Regularly Interspaced Short Palindromic Repeat (CRISPR)-Cas9-generated *Cabs1* and *AntiCabs1* KO mice. (**B**,**C**) Validation of the *Cabs1* KO mouse model by Western blot analysis. The loss of the Cabs1 protein expression in the testis and sperm confirmed the *Cabs1* KO strategy. (**D**) Confirmation of the *Cabs1* KO mouse model by immunofluorescence on testicular tissue.

**Figure 3 ijms-22-00652-f003:**
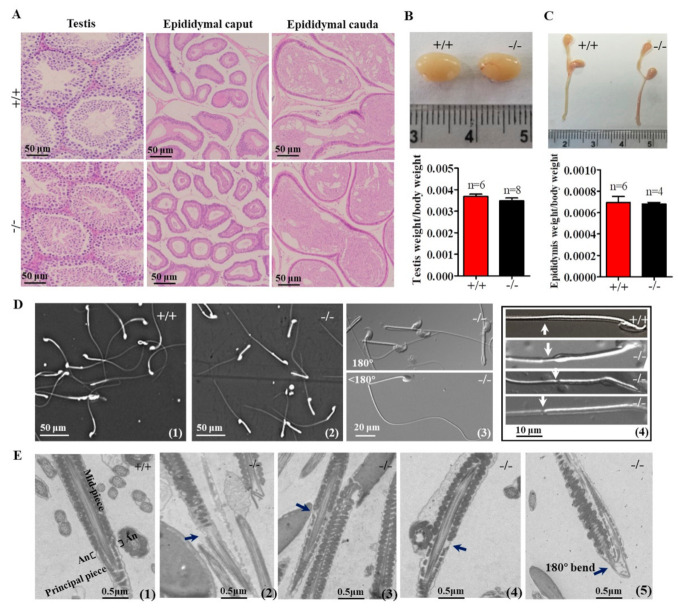
Testicular, epididymal, and spermatozoa phenotypes. (**A**) Histological analyses of the testis and epididymis sections from wild-type (WT) and calcium-binding protein spermatid-associated 1 knockout (*Cabs1*^−/−^) mice stained with hematoxylin and eosin. (**B**,**C**) Representative pictures and weights of testes and epididymites from WT and *Cabs1*^−/−^ mice. (**D**) Phase contrast images of spermatozoa collected from the cauda epididymis of adult WT and *Cabs1*^−/−^ mice. Spermatozoa from *Cabs1*^−/−^ mice had a tail bent at the midpiece–principal piece junction and a thinning of the annulus (white arrows). (**E**) Ultrastructural analysis (via transmission electron microscopy) of sperm flagella from WT and *Cabs1*^−/−^ mice. Flagella sections of normal and mutant sperm showing the thinning of annulus and tail bending (black arrows). (**1**) The annulus of sperm flagella from WT mice. (**2**–**4**) Deficient annulus of sperm flagella from *Cabs1*^−/−^ mice. (**5**) Bent tail.

**Figure 4 ijms-22-00652-f004:**
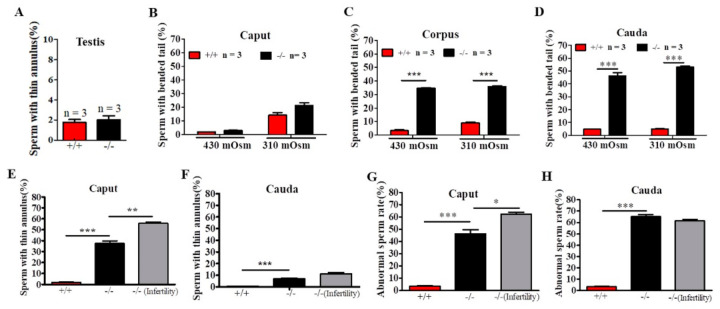
The site of the onset of structural defects of sperm flagella in calcium-binding protein spermatid-associated 1 knockout (*Cabs1*^−/−^) mice. (**A**–**C**) Enumeration of angulated spermatozoa in the three distinct regions of the epididymis indicated that the structural defect of bent tails occurred during the transit of sperm through the *Cabs1*^−/−^ epididymis. Sperm with bent tails were counted under 430 and 310-mOsm high-salt buffers. (**D**) The percentage of sperm with a thin annulus in the testes in WT *Cabs1*^−/−^ mice. (**E**,**F**) The percentage of sperm with a thin annulus in the caput and cauda of epididymis were calculated in WT and fertile and infertile *Cabs1*^−/−^ mice. (**G**,**H**) Total abnormal sperm rates in the caput and cauda of the epididymis were calculated in WT and fertile and infertile *Cabs1*^−/−^ mice. Data in bar graphs are presented as mean ± SE. * *p* < 0.05, ** *p* < 0.01, and *** *p* < 0.001.

**Figure 5 ijms-22-00652-f005:**
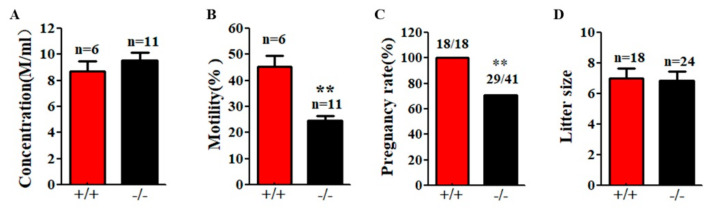
Motility and function defects of sperm flagella in calcium-binding protein spermatid-associated 1 knockout (*Cabs1*^−/−^) mice. (**A**) Sperm concentration and (**B**) motility in male *Cabs1*^−/−^ and wild-type (WT) mice. Sperm collected from the cauda epididymitis were measured by computer-assisted sperm analysis. (**C**) Male fertility and (**D**) litter size of adult *Cabs1*^−/−^ and WT mice. Pregnancy and litter size were examined in mating cages comparing male *Cabs1*^−/−^ and WT mice over a period of 3 months. Data in bar graphs are presented as mean ± SE. ** *p* < 0.01.

**Figure 6 ijms-22-00652-f006:**
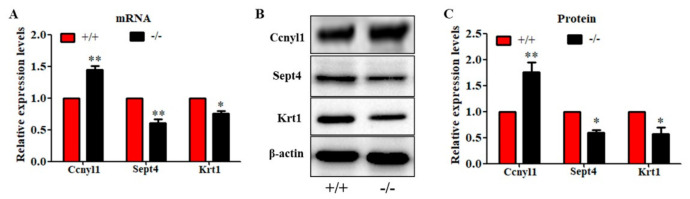
Genetic loss of calcium-binding protein spermatid-associated 1 (Cabs1) altered the expression levels of cytoskeleton-related proteins in sperm. (**A**) mRNA, (**B**,**C**) protein expression levels of cyclin Y-like 1 (*Ccnyl1*), septin 4 (*Sept4*), and *Krt1* in WT and KO mice. Total RNA was isolated from sperm using a RNAiso Plus kit, and qPCR was performed to detect mRNA expression levels. *β-actin* gene expression was used as a loading control. Equal amounts of total protein extracted from sperm of *Cabs1*^−/−^ and wild-type mice were subjected to SDS-PAGE and immunoblotting with antibodies directed against Ccnyl1, Sept4, and Krt1 and β-actin. (**B**) Blots are representative of three independent experiments, and the combined data are presented in histograms (**C**). Data in bar graphs are presented as mean ± SE. * *p* < 0.05 and ** *p* < 0.01 versus WT.

**Figure 7 ijms-22-00652-f007:**
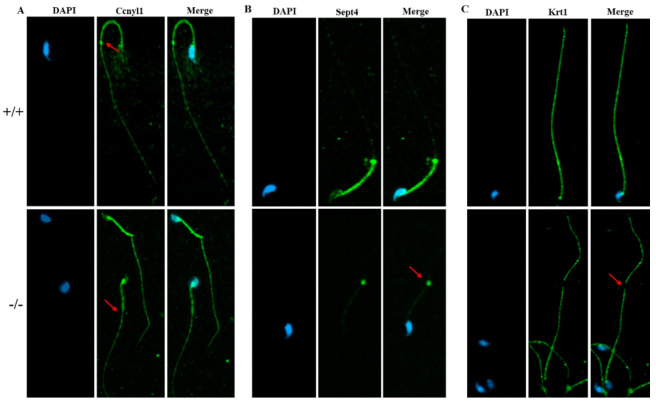
The expression of Ccnyl1, Sept4, and Krt1 (**A**–**C**) in calcium-binding protein sperma-tid-associated 1 knockout (Cabs1^−/−^) mice. Immunofluorescence labeling of mature sperm with cy-clin Y-like 1 (Ccnyl1), septin 4 (Sept4), and Krt1 antibodies, and nuclear staining with DAPI (4′,6-Diamidino-2-Phenylindole, Dihydrochloride). Imaging was performed on a confocal micro-scope. Arrows indicate a protein loss in KO mice sperm flagella compared with WT.

## Data Availability

Data is contained within the Appendix A.

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
