# Peer review of "Cabs1 Maintains Structural Integrity of Mouse Sperm Flagella during Epididymal Transit of Sperm"

_ijms, 2021, doi:10.3390/ijms22020652_

Round 1
Reviewer 1 Report
The manuscript „Cabs1 maintains structural integrity of sperm flagella in spermiogenesis“ with Zeng as corresponding author describes the CRISPR/Cas9 mediated generation and phenotypical characterization of Cabs1 and anti-Cabs1 deficient mice. The authors show that Cabs1 and anti-Cabs1 are mainly expressed in male reproductive organs. While anti-Cabs1 seems to be dispensable for male fertility, authors claim that Cabs1 deficiency results in sterility in approx. 30% of males. Authors obtain defects in flagellar differentiation, impaired motility and deregulated protein expression of cytoskeletal proteins in Cabs1 deficient mice claiming that Cabs1 is essential for assembly of the sperm flagellum.
In summary, the novelty of the submitted article is limited as a Cabs1 deficient mouse model has been published recently (Sun et al, 2020), showing that Cabs1 is dispensable for male fertility. This is partially in contrast to the present study and requires an in depth discussion and analysis. Although findings will be of interest for the scientific community, the manuscript requires several additional experiments and a differentiation between the sterile and fertile groups of Cabs1 deficient males. In addition, the acrosomal function of Cabs1 needs to be investigated. Further, many important information on the generated mouse lines require a more detailed description, which makes it difficult to evaluate the findings of the manuscript in its present form.
General comments:
- It is almost impossible to evaluate the quality of presented blots as all images are cropped and uncropped images (which are recommended by the journal to be included in the Supplement) are missing.
- Figure 1, Figure 3A-C, Figure 5D: The novelity of presented data is very limited as similar results have been published already (Sun et al., 2020).
- The authors claim that 30% of Cabs1 deficient mice are sterile whereas 70% show normal fecundity. This observation requires that both groups are analysed separately. In the present study (Fig. 2 B-C, Fig.3, Fig. 4 A-D, Figure 5A-B, Figure 6) authors do not explain which mice have been used (sterile mice, fertile mice, a mix of both?). How do authors explain the different phenotypes of Cabs1 deficient mice?
- The title of the manuscript is misleading. Authors show that observed defects seem to arise during epididymal transit of sperm but not during spermiogenesis. This should be reflected in the title of the manuscript. Otherwise, additional experiments analyzing testicular sperm of Cabs1 deficient mice are required (e.g. electron microscopy, western blot on testicular lysates for cytoskeletal proteins).
- The authors show that Cabs1 is expressed in the flagellum as well as in the acrosome of sperm. However, the study solely focusses on the effects of Cabs1 deficiency on flagellar differentiation and function. What is the effect of Cabs1 deficiency on acrosomal function? Without analyzing the effects on acrosomal function, the claim that impaired function of the flagellum is the underlying cause of male sterility cannot be made.
- Authors decided to analyze the protein expression of Cnyl1, TAT1, Sept4, Cytokeratin1 and a-actin. Please describe why these proteins have been chosen for analysis. What is the function of these proteins? Is there any information in literature that these proteins interact with CABS1? Are differences in protein level only present in epididymal sperm or do they arise in the testis already?
- Which animals were used for analysis? F1 generation? Usually, CRISPR/Cas9 mediated genome editing results in mosaic founder animals with several differentially gene-edited alleles. Did the authors sequence the generated alleles and separate them by breeding? Which line was established and analyzed finally (please provide a description of the generated mouse lines according to common nomenclature).
- The genotyping strategy for Cabs1 deficient mice is not comprehensively described. Further, PCRs for antiCabs1 require optimization as lot of unspecific bands are produced (Figure A2 D).
- Applies to all microscopic images: please add scale bars
- Applies to the whole manuscript: please adhere to the guidelines of the HUGO gene nomenclature
Author Response
The manuscript “Cabs1 maintains structural integrity of sperm flagella in spermiogenesis” with Zeng as corresponding author describes the CRISPR/Cas9 mediated generation and phenotypical characterization of Cabs1 and anti-Cabs1 deficient mice. The authors show that Cabs1 and anti-Cabs1 are mainly expressed in male reproductive organs. While anti-Cabs1 seems to be dispensable for male fertility, authors claim that Cabs1 deficiency results in sterility in approx. 30% of males. Authors obtain defects in flagellar differentiation, impaired motility and deregulated protein expression of cytoskeletal proteins in Cabs1 deficient mice claiming that Cabs1 is essential for assembly of the sperm flagellum.
In summary, the novelty of the submitted article is limited as a Cabs1 deficient mouse model has been published recently (Sun et al, 2020), showing that Cabs1 is dispensable for male fertility. This is partially in contrast to the present study and requires an in depth discussion and analysis. Although findings will be of interest for the scientific community, the manuscript requires several additional experiments and a differentiation between the sterile and fertile groups of Cabs1 deficient males. In addition, the acrosomal function of Cabs1 needs to be investigated. Further, many important information on the generated mouse lines require a more detailed description, which makes it difficult to evaluate the findings of the manuscript in its present form.
Answer: Thank you for your constructive suggestions. Since this paper was published very recently and the word “Cabs1” was not in the title and abstract, we didn't pay attention to it during the preparation and submission of our manuscript. Sorry for this neglection. In fact, they found that 13 testis- or epididymis-enriched genes including Cabs1 are dispensable for male infertility. Regarding Cabs1, the major information is from Table 1, showing that Cabs1 KO mice exhibited similar averaged litter size as wild type mice. However, the number of litters from Cabs1 KO mice was close to half of that from wild type mice (13 vs 30), suggesting a subfertility of Cabs1 KO in their study. Although there was no information whether Cabs1 KO mice could be separated as fertile and sterile groups in that study, their fertility result appears consistent with our results in Figure 5D. Except for the fertility test, other reproductive phenotypes including the testicular and epididymal development, sperm morphology, concentration and motility had not been addressed. In our study, we mainly focused on changes of sperm morphology and further investigated the places and causes of deformities. Our study demonstrates that deformities of sperm mainly occur during the normal transit of spermatozoa in the epididymis and Cabs1 is an important component of the sperm annulus essential for proper sperm tail assembly and motility. The roles of Cabs1 in sperm function such as capacitation, acrosome reaction and fertility will be further explored in our subsequent study. The information of the generated mouse lines were provided in Figure S6.
General comments:
It is almost impossible to evaluate the quality of presented blots as all images are cropped and uncropped images (which are recommended by the journal to be included in the Supplement) are missing.
Answer: Thank you for your advice. The intact images are listed in the supplement (Figure S5).
Figure 1, Figure 3A-C, Figure 5D: The novelty of presented data is very limited as similar results have been published already (Sun et al., 2020).
Answer: We really appreciate that you provide this new information. However, as explained above, Sun’s paper only presented the fertility data from 3 male mice, which is generally consistent with our fertility result. Other reproductive phenotypes regarding the testicular and epididymal development, sperm morphology, concentration and motility hadn’t been addressed in Sun et al paper.
The authors claim that 30% of Cabs1 deficient mice are sterile whereas 70% show normal fecundity. This observation requires that both groups are analyzed separately. In the present study (Fig. 2 B-C, Fig.3, Fig. 4 A-D, Figure 5A-B, Figure 6) authors do not explain which mice have been used (sterile mice, fertile mice, a mix of both?). How do authors explain the different phenotypes of Cabs1 deficient mice?
Answer: Because we mainly focused on the changes of sperm morphology in the present study, only deformity differences were analyzed and counted both the sterile and fertile KO mice (Fig.4). As for the causes and mechanisms of Cabs1 deletion induced subfertility, we are conducting in-depth analysis. If there is no specific note, the data are from a mix of both mice. The different phenotypes of Cabs1 deficient mice might be caused by sperm function dysregulation reaching to maximal threshold. It has been reported that Cabs1 involve in acrosome reaction and phosphorylation regulation, which are essential for sperm function. The related discussion has been incorporated into the revision (p8, line238-243).
The title of the manuscript is misleading. Authors show that observed defects seem to arise during epididymal transit of sperm but not during spermiogenesis. This should be reflected in the title of the manuscript. Otherwise, additional experiments analyzing testicular sperm of Cabs1 deficient mice are required (e.g. electron microscopy, western blot on testicular lysates for cytoskeletal proteins).
Answer: We appreciate you pointing out this inappropriate expression. We have revised our title to “Cabs1 maintains structural integrity of mouse sperm flagella during epididymal transit of sperm”
The authors show that Cabs1 is expressed in the flagellum as well as in the acrosome of sperm. However, the study solely focuses on the effects of Cabs1 deficiency on flagellar differentiation and function. What is the effect of Cabs1 deficiency on acrosomal function? Without analyzing the effects on acrosomal function, the claim that impaired function of the flagellum is the underlying cause of male sterility cannot be made.
Answer: Thank you. It is a good comment. We are performing the related experiments to investigate the roles of Cabs1 in acrosome. This will help us uncover the function of Cabs1 in mature sperm and the mechanisms of Cabs1 deletion caused-subfertility. The impaired flagellum is one cause of infertility. We supplemented the related discussion in our revision (p8, line 238-245).
Authors decided to analyze the protein expression of Cnyl1, TAT1, Sept4, Cytokeratin1 and a-actin. Please describe why these proteins have been chosen for analysis. What is the function of these proteins? Is there any information in literature that these proteins interact with CABS1? Are differences in protein level only present in epididymal sperm or do they arise in the testis already?
Answer: It’s a good comment. Cnyl1, TAT1 and Sept4 are essential for sperm flagellum assembly, especially for the annulus formation (Touré et al., 2007; Zi et al., 2015 and Kissel et al., 2005). Their KO mice have similar phenotypes with Cabs1. Defect of them also resulted in an abnormal annulus and a bended tail. Cytokeratin1 and α-actin play crucial roles in maintaining cytoskeleton, which might be also involved in structural integrity of sperm flagella. Until now, there are no literatures that these proteins interact with Cabs1. Some of them also were dysregulated in the testis, however, in the present study we mainly focused on these protein changes in epididymal sperm. The intact WB results were shown in Figure S5.
Which animals were used for analysis? F1 generation? Usually, CRISPR/Cas9 mediated genome editing results in mosaic founder animals with several differentially gene-edited alleles. Did the authors sequence the generated alleles and separate them by breeding? Which line was established and analyzed finally (please provide a description of the generated mouse lines according to common nomenclature).
Answer: Yes, you are right. Three lines F1 generation KO mice were produced and bred separately. Only one line of KO mice was retained for the next experiments because they have similar phenotypes in sperm morphology. The details were provided in Figure S6.
The genotyping strategy for Cabs1 deficient mice is not comprehensively described. Further, PCRs for antiCabs1 require optimization as lot of unspecific bands are produced (Figure A2 D).
Answer: Thank you for your suggestion. The genotyping of Cabs1 mice were conducted by PCR followed sequencing. The detailed information was provided in revised manuscript (Figure S6). Because AntiCabs1 transcript has many highly overlapped regions with Cabs1, so the design of primers is greatly restricted resulting in some unspecific bands.
Applies to all microscopic images: please add scale bars
Applies to the whole manuscript: please adhere to the guidelines of the HUGO gene nomenclature
Answer: Thanks. All these information are added or revised. For the AntiCabs1, there is no official symbol until now, so we added its gene ID from the NONCODE database (p9, line 302).
Reviewer 2 Report
This study analyzed the role of Cabs1 and its antisense strand ncRNA in mouse spermatogenesis by generating knockdown animals using CRISPR-Cas9 genome editing. The role of Cabs1 in spermatogenesis is largely unknown, and this study will provide important insights. The data have no problem to speculate the conclusions. Although there is no uncertainty about the results of the experiments, it does not seem appropriate to discuss why Cabs1-/- male mice show abnormalities in sperm morphology yet maintain reduced fertility. Therefore, the discussion will need to be elaborated.
Major points
1) It should be noted that the data in Fig. 4 and Fig. 5A and B are from epididymal spermatozoa, whereas Fig. 5C and D are the result of mating. The mating results indicate that epididymal and ejaculated spermatozoa may have different fertility properties. The authors should clarify this point and revise their discussion. The uterus and fallopian tubes are thought to eliminate faulty sperm. In this sense, the authors could perform IVF and consider the fertility of spermatozoa with abnormal morphology.
2) With regard to the experiment in Fig. 6, what results do you get in terms of mRNA expression level, although you are examining it in terms of protein levels? It is also necessary to show by immunostaining whether the original localization of the three with increased or decreased protein levels is preserved.
Minor points
1)I want the word "mouse" in the title.
2)Please check the spell “prostrate” in line 210.
Author Response
Comments and Suggestions for Authors
This study analyzed the role of Cabs1 and its antisense strand ncRNA in mouse spermatogenesis by generating knockdown animals using CRISPR-Cas9 genome editing. The role of Cabs1 in spermatogenesis is largely unknown, and this study will provide important insights. The data have no problem to speculate the conclusions. Although there is no uncertainty about the results of the experiments, it does not seem appropriate to discuss why Cabs1-/- male mice show abnormalities in sperm morphology yet maintain reduced fertility. Therefore, the discussion will need to be elaborated.
Answer: Thank you very much. It is a good comment. In fact, why part of Cabs1 KO mice maintained fertility while others were sterile also puzzles us. For the mice showing normal fertility, we speculate that the remaining morphologically normal sperm is enough to fertilize the eggs. On the other hand, for the sterile mice, Cabs1 might serve as a regulator for other sperm functions such as acrosome reaction which contributes to a threshold value which might be critical to maintain the fertility. Nevertheless, further experiments are definitely required to test the above hypothesis and explore the underlying mechanisms. This information has been incorporated in the Discussion Section. (p8, line 238-243)
Major points
1) It should be noted that the data in Fig. 4 and Fig. 5A and B are from epididymal spermatozoa, whereas Fig. 5C and D are the result of mating. The mating results indicate that epididymal and ejaculated spermatozoa may have different fertility properties. The authors should clarify this point and revise their discussion. The uterus and fallopian tubes are thought to eliminate faulty sperm. In this sense, the authors could perform IVF and consider the fertility of spermatozoa with abnormal morphology.
Answer: Thank you for your suggestions. Clarifying whether epididymal and ejaculated sperm have different fertility properties and whether sperm with abnormal morphology still keeps fertility in IVF will definitely help understanding this study. We will perform the suggested experiments to investigate this issue. Additional discussion has been added in our revision (p8, line 232-233).
2) With regard to the experiment in Fig. 6, what results do you get in terms of mRNA expression level, although you are examining it in terms of protein levels? It is also necessary to show by immunostaining whether the original localization of the three with increased or decreased protein levels is preserved.
Answer: Thank you for your advice. Given that sperm are terminal differentiated cells, they are transcriptional silencing or hardly transcriptional in the epididymis. So we just investigated the related gene expression in protein levels. The co-localization and interaction of Cabs1 with these proteins are conducting.
Minor points
1)I want the word "mouse" in the title.
2)Please check the spell “prostrate” in line 210.
Answer: We appreciate you pointing out these inappropriate expressions. They have been corrected in our revision.
Reviewer 3 Report
Calcium binding protein Cabs1 is a spermatid-specific protein. The cellular localization has been already revealed in other papers. Authors performed knockout experiments to reveal the function of the protein. Although the knockout of AntiCabs1 showed no phenotype, Cabs1 knockout mice had a structural defect in sperm flagellar leading subfertility. This defect progresses with the maturation of spermatozoa in the epididymis. The expressions of several related proteins were changed in sperm. The detail mechanism by which the defect was caused in absent of Cabs1 is still almost unknown. This is first report showing that systemic deletion of Cabs1 causes sperm-specific phenotype.
The experiments were well designed and performed. The data was clearly shown. The discussion is reasonable. Overall, it is highly complete and there are no doubts.
Author Response
Calcium binding protein Cabs1 is a spermatid-specific protein. The cellular localization has been already revealed in other papers. Authors performed knockout experiments to reveal the function of the protein. Although the knockout of AntiCabs1 showed no phenotype, Cabs1 knockout mice had a structural defect in sperm flagellar leading subfertility. This defect progresses with the maturation of spermatozoa in the epididymis. The expressions of several related proteins were changed in sperm. The detail mechanism by which the defect was caused in absent of Cabs1 is still almost unknown. This is first report showing that systemic deletion of Cabs1 causes sperm-specific phenotype.
The experiments were well designed and performed. The data was clearly shown. The discussion is reasonable. Overall, it is highly complete and there are no doubts.
Answer:We really appreciate your positive comments and encouragement. In fact, the other reviewer raised some points for further discussion, and we input our thoughts in the revised manuscript. Thanks again.
Reviewer 4 Report
In this study, Zhang et al. , indicated that male reproductive function of cabs1 and Cabs1 gene antisense in vivo through kncokout mice model. The major defects of cabs1 null mice exhibited significantly impaired structure of sperm-tail and revels subfertility. And, the major defect of sperm-tail is with annulus, which may explain the phenotype of sperm with a bent tail. For dissect the possible reason of the causing male infertility, they found that increased septin 4 and cytokeratin 1 and decreased cyclin Y-like 1 levels in cabs1 null mice, compared with wild-type. However, no any phenotype form Cabs1 gene antisense knockout mice. This study demonstrates that Cabs1 is essential for the integration of the sperm annulus.
Several points:
- please provide the Approved no. and date of Animal study.
- More Introduction and discussion of function domain of Cabs1 in section of Introduction and discussion are needed to link the phenotype.
- It is hard to link the phenotype of cabs1 in mice to the decreased cnyl1 and tat1, could you provide any model or explain in the section of discussion ?
- The phenotype of antisense cabs1 KO mice may assign as "supplement results " instead of the section of appendix.
Author Response
In this study, Zhang et al. , indicated that male reproductive function of cabs1 and Cabs1 gene antisense in vivo through kncokout mice model. The major defects of cabs1 null mice exhibited significantly impaired structure of sperm-tail and revels subfertility. And, the major defect of sperm-tail is with annulus, which may explain the phenotype of sperm with a bent tail. For dissect the possible reason of the causing male infertility, they found that increased septin 4 and cytokeratin 1 and decreased cyclin Y-like 1 levels in cabs1 null mice, compared with wild-type. However, no any phenotype form Cabs1 gene antisense knockout mice. This study demonstrates that Cabs1 is essential for the integration of the sperm annulus.
Several points:
1. Please provide the Approved no. and date of Animal study.
Answer:Thanks for pointing this out. This information has been provided in our revision (p9, line313).
2. More Introduction and discussion of function domain of Cabs1 in section of Introduction and discussion are needed to link the phenotype.
Answer:Thank you for your advice. The related information was discussed in our revision (p2, line 55-57; p8, line 239-244).
3. It is hard to link the phenotype of cabs1 in mice to the decreased cnyl1 and tat1, could you provide any model or explain in the section of discussion?
Answer:Cnyl1, Tat1 KO mice have similar phenotypes with Cabs1 (Touré et al., 2007 and Zi et al., 2015). They are essential for sperm flagellum assembly, especially for the annulus formation. So we investigated their expressions in Cabs1 KO mice. They might form a complex to assemble the annulus or have regulatory relationships (p8, line 280-287). We have added the discussion about this issue. Thank you very much.
4. The phenotype of antisense cabs1 KO mice may assign as "supplement results" instead of the section of appendix.
Answer:Thank you for your suggestion. We will rearrange these results according to journal’s requirements.
Round 2
Reviewer 1 Report
Thank you for providing a revised version of the manuscript, which shows several improvements and a more detailed description of the methodologies, which are essential for understanding and evaluation of the manuscript. In addition, the recently published mouse model from Sun et al. is discussed. However, the provided uncropped western blots are not convincing in a way that all antibodies, which were used in the study, generated multiple unspecific bands, sometimes with similar deregulations in band intensity as observed for the protein of interest. In conclusion, additional experiments are absolutely required to validate the oberserved changes in cytoskeletal structure, e.g. immunofluorescent visualization of the sperm cytoskeleton. The authors describe that cytoskeletal proteins were chosen because the knockout phenotype of these proteins show a similar phenotype as Cabs1 deficient mice. However, for 2 out of the 3 significantly deregulated proteins, authors observe an upregulation (Sept4 and Krt1) in Cabs1 knockout versus wildtype mice. Does this deregulation have a physiological relevance? Further, the additional experiments, which have been suggested in the first round of revision, would significantly increase the quality of the manuscript.
Author Response
Answer: Thank you for your suggestion. It is difficult to order many reagents including antibodies in this tough period, so we spent too long time to revise our manuscripts. Indeed, some new WB results are not inconsistent with previous, we use two different antibodies to confirm our data. Additionally, immunofluorescence results also support our conclusion. As components of the sperm ring structure, loss of function of Sept4 or Ccnyl1 caused sperm malformation and fertility. Here, Cabs1 deficiency led to changes of Sept4 or Ccnyl1 expression, which might contribute to sperm malformation and even subfertility. Expression level and pattern of Krt1 might cause the sperm tail structure to be easily damaged since Krt1 is one of cytoskeletal protein. Thanks again.
Reviewer 2 Report
It is disappointing that the additional experiments proposed were not performed, but I am satisfied with the revisions to the manuscript.
Minor points
1) I could not find the explanation about "(1), (2), (3), ..." in Figure 3D&E. What do these numbers indicate? If it's not mentioned, I think it should be deleted.
2) Please check the spell “tial” in line 288.
Author Response
It is disappointing that the additional experiments proposed were not performed, but I am satisfied with the revisions to the manuscript.
Answer: Thank you for your comments. We performed the additional experiments (IF) to improve our paper’s quality.
Minor points
1) I could not find the explanation about "(1), (2), (3), ..." in Figure 3D&E. What do these numbers indicate? If it's not mentioned, I think it should be deleted.
2) Please check the spell “tail” in line 288.
Answer: Thank you for your suggestion. They have been revised in our revision.